# Optimizing Management of Alfalfa (*Medicago sativa* L.) Nitrogen Fertilizer Based on Critical Nitrogen Concentration Dilution Curve Model

**DOI:** 10.3390/plants14121782

**Published:** 2025-06-11

**Authors:** Yaya Duan, Yi Ling, Haiyan Li, Wenjing Chang, Jiandong Lu, Minhua Yin, Yanxia Kang, Yanlin Ma, Yayu Wang, Guangping Qi, Guoyun Shen

**Affiliations:** 1College of Water Conservancy and Hydrpower Engineering, Gansu Agricultural University, Lanzhou 730070, China; 1073323020376@st.gsau.edu.cn (Y.D.); 1073324120786@st.gsau.edu.cn (Y.L.); 107332201100@st.gsau.edu.cn (H.L.); 1073324120789@st.gsau.edu.cn (W.C.); 1073324120819@st.gsau.edu.cn (J.L.); mayl@gsau.edu.cn (Y.M.); wangyy@gsau.edu.cn (Y.W.); qigp@gsau.edu.cn (G.Q.); 20212201028@st.gsau.edu.cn (G.S.); 2Gansu Province Jingtai Chuan Power Irrigation Water Resource Utilization Center, Baiyin 730400, China

**Keywords:** critical nitrogen concentration, nitrogen nutrition diagnostics, nitrogen application rates, alfalfa

## Abstract

The critical nitrogen dilution curve (CNDC) model enables precise nitrogen management by quantifying the threshold of nitrogen deficiency in crops, thereby enhancing both crop productivity and nitrogen use efficiency. However, its applicability to perennial crops remains unclear. In this study, alfalfa (*Medicago sativa* L.), a perennial leguminous forage, was used as the model crop. Based on two years of field experiments, CNDC models of aboveground biomass were constructed under two nitrogen fertilizer regimes: urea (0, 80, 160, and 240 kg·ha^−1^, applied in a 6:2:2 basal-to-topdressing ratio) and controlled-release urea (CRU; 0, 80, 160, and 240 kg·ha^−1^, applied as a single basal dose). Using these models, the nitrogen nutrition index (NNI) and cumulative nitrogen deficit (N_and_) models were developed to diagnose alfalfa nitrogen status, and the optimal nitrogen application rates were determined via regression analysis. The results showed that critical nitrogen concentration and aboveground biomass followed a power function relationship under both fertilizer types. For CRU treatments, parameters a and b were 3.41 and 0.20 (first cut), 3.15 and 0.12 (second cut), and 2.24 and 0.40 (third cut), respectively. For urea treatments, a and b were 3.13 and 0.35 (first cut), 2.21 and 0.16 (second cut), and 1.75 and 0.73 (third cut). The normalized root mean square error (n-RMSE) of the models ranged from 3.1% to 13%, indicating high model reliability. Based on the NNI, N_and_, and yield response models, the optimal nitrogen application rates were 175.44~181.71 kg·ha^−1^ for urea and 145.63~153.46 kg·ha^−1^ for CRU, corresponding to theoretical maximum yields of 14.76~17.40 t·ha^−1^ and 16.76~20.66 t·ha^−1^, respectively. Compared to urea, CRU reduced nitrogen input by 18.41~20.47% while achieving equivalent or higher theoretical yields. This study provides a scientific basis for nitrogen status diagnosis and precision nitrogen application in alfalfa cultivation.

## 1. Introduction

N is an essential nutrient for crop growth, and its use efficiency directly influences the sustainability of agricultural production and environmental outcomes [1,2]. The application of exogenous nitrogen fertilizers plays a vital role in improving soil fertility, maintaining nutrient balance, and promoting crop growth [3,4]. Globally, the annual nitrogen input to agricultural land reaches 200 million tons. However, due to extensive and inefficient management, nitrogen use efficiency remains below 50%, with substantial nitrogen losses through leaching and volatilization. These losses contribute to soil acidification, groundwater contamination, and greenhouse gas emissions, while also increasing agricultural production costs [5,6,7]. Inappropriate nitrogen management can also lead to metabolic imbalances in crops, weakening stress resistance and impairing the synthesis of proteins, amino acids, and vitamins, thereby reducing yield and quality [8]. Ensuring a balance between nitrogen supply and crop demand throughout the growth period is key to addressing these issues [9,10]. Therefore, real-time monitoring of nitrogen status in both plants and soils, as well as implementing a demand-driven nitrogen supply strategy, are critical for achieving the goal of “reduced nitrogen input, improved efficiency, high yield, and high quality” in sustainable agricultural production.

Real-time diagnosis of crop nitrogen nutrition is the foundation of rational nitrogen fertilization and plays a crucial role in balancing N supply and demand in the soil–plant system [11,12]. Current studies on crop N status monitoring mainly focus on two aspects: plant physiological responses and soil nitrogen supply. Plant-based approaches involve measurements such as leaf chlorophyll content, nitrogen accumulation, and vegetation indices derived from remote sensing. These methods aim to directly assess the plant’s nitrogen content or related physiological traits to indicate nitrogen uptake and utilization. However, their accuracy is often affected by factors such as leaf thickness, age gradient, and environmental stress, limiting their universality [13,14]. Emerging non-destructive remote sensing technologies, including multispectral and hyperspectral inversion models, estimate canopy nitrogen distribution based on near-infrared reflectance. While promising, these techniques are sensitive to climatic conditions, canopy structure, and varietal differences, and they often require high technical expertise and financial investment [15,16]. Soil is the source of nitrogen supply [17], and the existing monitoring of soil nitrogen nutrition mainly includes root zone nutrient (nitrate nitrogen, ammonium nitrogen, and total nitrogen content) monitoring, the 15N isotope tracer method, and nitrogen mineralization kinetic modeling [18]. These techniques effectively assess soil nitrogen supply capacity but have limitations in dynamic monitoring, equipment complexity, and the use of radioactive materials. The critical nitrogen concentration (N_c_) refers to the minimum nitrogen concentration required to achieve maximum biomass at a given growth stage [19]. It decreases with increasing biomass and typically follows a power-law relationship [19]. When diagnosing plant nitrogen nutrition based on the critical nitrogen concentration dilution curve model, the deviation between the actual nitrogen concentration of the plant and the critical nitrogen concentration can be compared in real time, enabling dynamic diagnosis of crop nitrogen nutrition status and prediction of nitrogen requirements [20]. This method has good universality and continuity. In recent years, the critical nitrogen dilution curve (CNDC) model has been widely applied to various crops to monitor plant nitrogen status. These include grain crops like rice [21], wheat [22], and maize [23]; forage crops such as tobacco [24], ryegrass [25], and sugarcane [26]; and root/tuber crops like carrot [27], potato [28], and garlic [29], and the standardized root mean square error values of the critical nitrogen concentration dilution curve model are all below 20%, demonstrating high accuracy in plant nutrient diagnosis. Furthermore, it was shown that the CNDC modeling also breaks through the limitation of single-parameter diagnosis and can be constructed based on organs such as leaves, leaf sheaths, and stems. Parameters such as chlorophyll content and leaf area, respectively, can adapt more flexibly to the growth characteristics of different crops and the nutritional diagnosis needs of different growth stages, and it has become an economical and effective tool for real-time diagnosis of plant nitrogen [21,22,23,24,25,26,27,28,29], making it an effective and economical tool for real-time plant nitrogen diagnosis.

To date, most CNDC studies have focused on annual grain and economic crops [30], while research on forage species, particularly perennial ones, such as Alfalfa, remains limited. Alfalfa (*Medicago sativa* L.), the most widely cultivated forage crop globally, is known as the “king of forages” due to its high nutritional value and strong stress resistance [31,32]. As a leguminous species with a deep root system, high leaf biomass, and strong soil cover, alfalfa significantly enhances soil fertility and reduces erosion, offering both productive and ecological benefits [33]. Located in northwest China, Gansu Province has abundant solar radiation and large diurnal temperature variation, making it ideal for alfalfa cultivation. In recent years, Gansu has become a major production base for high-quality alfalfa, accounting for over 60% of China’s commercial planting area [34]. However, challenges such as arid climate and poor soils limit production potential [35]. In this context, the present study systematically analyzed nitrogen accumulation dynamics in alfalfa across different harvests under applied urea and controlled-release nitrogen fertilizer. The objectives were to (1) quantify differences in biomass accumulation and plant nitrogen concentration under the two nitrogen types and construct CNDC models; (2) develop nitrogen nutrition index (NNI) and cumulative nitrogen deficit (N_and_) models based on CNDC to diagnose nitrogen surplus or deficiency thresholds throughout the growing season; and (3) evaluate the yield-increasing and nitrogen-saving potential of controlled-released urea (CRU) in arid regions of Northwest China to provide a theoretical basis for precision nitrogen management in alfalfa production.

## 2. Results and Analysis

### 2.1. Modeling of Critical Nitrogen Concentration Dilution Curves in Alfalfa

#### 2.1.1. Effect of Nitrogen Fertilizer Management on Alfalfa Aboveground Biomass

Nitrogen fertilizer management had a significant effect on the aboveground biomass of alfalfa (*p* < 0.05, Figure 1). Overall, aboveground biomass followed the order: first cut > second cut > third cut, and it increased with rising nitrogen application rates. Under urea treatment, the aboveground biomass of alfalfa ranged from 4.79 to 9.03 t·ha^−1^. Compared to N0, N1, N2, and N3 increased biomass by 5.24–30.68% (first cut), 4.16–28.82% (second cut), and 4.07–27.72% (third cut), respectively. Under controlled-release urea (CRU) treatment, aboveground biomass ranged from 4.79 to 10.36 t·ha^−1^, with C1, C2, and C3 increasing biomass by 15.55–50.14% (first cut), 13.45–42.37% (second cut), and 11.27–39.56% (third cut), respectively, compared to C0. At the same nitrogen application level, CRU consistently resulted in 6.71–14.84% higher aboveground biomass than urea. In the mid to late stages of alfalfa growth, biomass accumulation rates of N2 and C2 exceeded those of N3 and C3, indicating a plateauing trend at higher nitrogen inputs. The biomass response followed statistically significant inequalities for the two nitrogen types: urea: N0 < N1 < N2 ≈ N3, CRU: C0 < C1 < C2 ≈ C3.

#### 2.1.2. Effects of Nitrogen Fertilizer Management on Alfalfa Plant Nitrogen Concentration

The nitrogen concentration in alfalfa plants showed a decreasing trend as the growth period progressed (Figure 2), indicating a nitrogen dilution effect during biomass accumulation. With increasing nitrogen application rates, the nitrogen concentration increased correspondingly. At the same nitrogen rate, CRU resulted in higher nitrogen concentrations in alfalfa plants compared to urea (Table 1). Under urea treatment, nitrogen concentrations across the three cuts ranged from 2.24 to 4.61% (Figure 2a), 1.97 to 3.93% (Figure 2b), and 1.72 to 3.61% (Figure 2c), respectively. Significant differences were observed between N2 and N0, N3 and N0, and N3 and N1. Under CRU treatment, nitrogen concentrations ranged from 2.24 to 4.62% (Figure 2d), 1.97 to 4.16% (Figure 2e), and 1.72 to 3.83% (Figure 2f), respectively. Significant differences were found between C2 and C0, C3 and C0, and C3 and C1.

#### 2.1.3. Construction of Critical Nitrogen Dilution Curve Models for Alfalfa

Critical nitrogen dilution curve models were separately constructed for the aboveground biomass of alfalfa under urea and CRU treatments across the three cuts (Figure 3). The models showed excellent fit, with all coefficients of determination (R^2^) exceeding 0.99, indicating that the established models effectively described the relationship between aboveground biomass and critical nitrogen concentration. For the CRU treatments, the parameter a values for the first, second, and third cuts were 3.41, 3.15, and 2.24, respectively, all higher than those under urea treatments (3.13, 2.21, and 1.75). This suggests that CRU improved the nitrogen uptake capacity in alfalfa plants compared to urea. In contrast, the b values under CRU treatment (0.20, 0.12, and 0.40) were lower than those under urea (0.35, 0.16, and 0.73), indicating that nitrogen concentration in alfalfa plants decreased more slowly with biomass accumulation under CRU than under urea, i.e., a slower nitrogen dilution rate.

### 2.2. Validation of the Critical Nitrogen Dilution Curve Model for Alfalfa

The critical nitrogen dilution models constructed in 2023 were validated using experimental data from the 2024 growing season. During the 2024 season, the maximum aboveground biomass of alfalfa under urea treatment for the first, second, and third cuts ranged from 6.9 to 8.4 t·ha^−1^, 5.7 to 6.9 t·ha^−1^, and 5.3 to 6.4 t·ha^−1^, respectively. Under CRU treatment, the maximum aboveground biomass for the three cuts ranged from 7.0 to 9.3 t·ha^−1^, 5.9 to 7.5 t·ha^−1^, and 5.5 to 6.9 t·ha^−1^, respectively. These maximum biomass values were substituted into the respective critical nitrogen dilution equations to calculate the simulated critical nitrogen concentrations. The simulated values were then compared with measured values through regression analysis (Figure 4). The resulting coefficients of determination (R^2^) ranged from 0.96 to 0.99, and the normalized root mean square error (n-RMSE) ranged from 3.1 to 13%. These results indicate that the critical nitrogen dilution models for alfalfa under both urea and CRU treatments exhibit high stability and accuracy.

### 2.3. Nitrogen Nutrition Diagnosis of Alfalfa Plants Under Different Nitrogen Fertilizer Management Strategies

#### 2.3.1. Nitrogen Nutrition Diagnosis Based on NNI

The NNI of alfalfa plants increased with rising nitrogen application rates and decreased as the growth period progressed (Figure 5). During the alfalfa growth cycle, the NNI values of N0, N1, C0, and C1 consistently remained below 1, indicating that these nitrogen application rates inhibited nitrogen uptake and biomass accumulation. In contrast, the NNI values of N3 and C3 were consistently above 1, suggesting that these application levels resulted in nitrogen surplus. The NNI value of N2 was slightly below 1, while that of C2 was slightly above 1. These results indicate that the optimal nitrogen application rate for alfalfa is slightly above 160 kg·ha^−1^ when using urea, and slightly below 160 kg·ha^−1^ when using CRU.

To further evaluate the feasibility of using the NNI to predict nitrogen surplus or deficiency in alfalfa plants, the relationship between NNI and relative yield (RY) was established (Figure 6). The coefficients of determination (R^2^) under both urea and CRU treatments reached highly significant levels, indicating a strong fit between NNI and RY and suggesting that NNI can be effectively used to predict nitrogen nutritional status in alfalfa. In addition, the NNI values corresponding to maximum RY under urea treatment were higher than those under CRU treatment, indicating that higher nitrogen application is required to achieve maximum yield when using urea compared to CRU.

#### 2.3.2. Nitrogen Nutrition Diagnosis of Alfalfa Based on N_and_

The N_and_ of alfalfa decreased with increasing nitrogen application rates and showed increasing variation among treatments as the growth period progressed (Figure 7). The N_and_ values ranged from −8.92 to 25.69 under urea treatment and from −8.71 to 24.56 under CRU treatment. Specifically, N0, N1, C0, and C1 had N_and_ > 0, indicating nitrogen deficiency in plant accumulation. In contrast, N3 and C3 had N_and_ < 0, indicating nitrogen surplus. For N2, N_and_ fluctuated around zero in the early growth stage but trended toward values greater than zero (N_and_ > 0) during the middle to late stages. For C2, N_and_ remained slightly below zero throughout the entire growth cycle. These results suggest that the optimal nitrogen application rate for alfalfa is slightly above 160 kg·ha^−1^ when using urea and slightly below 160 kg·ha^−1^ when using CRU. This finding is consistent with the diagnostic results based on the nitrogen nutrition index (NNI) described in Section 2.3.1.

Similar to NNI, the relationship between N_and_ and RY was established (Figure 8, *p* < 0.05). The coefficients of determination (R^2^) for both urea and CRU treatments were high, ranging from 0.902~0.994, indicating that the model effectively explains the relationship between N_and_ and RY. The quadratic term coefficient in the fitted curve reflects the sensitivity of alfalfa yield to nitrogen deficit. The absolute value of the quadratic coefficient under urea treatment was greater than that under CRU treatment, suggesting that yield in urea-treated alfalfa is more sensitive to nitrogen deficit fluctuations and thus requires more precise nitrogen management.

### 2.4. Effects of Nitrogen Fertilizer Management on Alfalfa Yield

Fitting alfalfa yield against nitrogen application rates for both fertilizer types (Table 2) revealed that the optimal nitrogen rate for urea ranged from 175.44 to 181.71 kg·ha^−1^, corresponding to maximum yields of 14.76 to 17.40 t·ha^−1^ and the optimal nitrogen rate for CRU ranged from 145.63 to 153.46 kg·ha^−1^, corresponding to maximum yields of 16.76 to 20.66 t·ha^−1^. These findings are consistent with the optimal nitrogen rates estimated using the NNI and N_and_ models.

## 3. Discussion

### 3.1. Critical Nitrogen Dilution Curve Model for Alfalfa

The essence of nitrogen dilution in crops lies in the fact that the rate of biomass accumulation exceeds the rate of nitrogen uptake or supply, resulting in a decline in nitrogen concentration per unit of biomass [20]. The use of the CNDC model to diagnose crop nitrogen status has been widely applied to various cereal and cash crops [36,37,38]. In the CNDC model, parameter a represents the nitrogen concentration when the aboveground biomass reaches 1 t·ha^−1^, while b represents the rate at which nitrogen concentration decreases with increasing biomass [39,40]. In this study, we developed CNDC models for alfalfa based on aboveground biomass under urea and CRU treatments. The fitted equations were as follows: Urea: N_c_ = 3.13W^−0.35^, N_c_ = 2.21W^−0.16^, N_c_ = 1.75W^−0.73^, CRU: N_c_ = 3.41W^−0.20^, N_c_ = 3.15W^−0.12^, N_c_ = 2.24W^−0.40^ (Figure 3). All parameter a values were lower than those reported by Lemaire et al. [41] for alfalfa (N_c_ = 4.8W^−0.34^), likely due to differences in climatic and soil nutrient conditions. Lemaire’s [41] experiments were conducted in France under mild and humid conditions with evenly distributed rainfall and favorable soil hydrothermal environments, which promoted root vigor, nitrogen uptake, and carbon–nitrogen metabolism in alfalfa. In contrast, our experiment was conducted in the arid climate of Northwest China, where low precipitation and frequent water stress reduced photosynthetic efficiency and carbon-nitrogen assimilation, resulting in lower nitrogen concentrations in alfalfa plants. However, Chen et al. [42] constructed a critical nitrogen concentration model for alfalfa: N_c_ = 3.352W^0.391^ (Aohan alfalfa) and N_c_ = 2.673W^0.485^ (Gongnong No.1 alfalfa); the model parameters of Chen et al.’s study differ significantly from those of the critical nitrogen concentration dilution curve model constructed in this study. This may be due to differences in the research objectives and the composition of aboveground biomass. Chen et al. [42] focused on seed-producing alfalfa, with the aboveground biomass of alfalfa including leaves, stems, flowers, pods, and seeds. This study focused on forage alfalfa, with the aboveground biomass of alfalfa comprising only leaves and stems.

As a perennial leguminous forage, alfalfa presents complex nitrogen dynamics due to factors such as biological nitrogen fixation, nitrogen redistribution after cutting, and root–rhizobia interactions [43,44]. Our findings showed that model parameters varied across fertilizer types and harvests (Figure 3). Under the same biomass level, CRU-treated plants exhibited higher parameter a values and lower b values compared to urea, indicating higher nitrogen concentration and slower dilution rates. This is attributed to the nutrient release characteristics of the fertilizers: CRU releases nitrogen gradually through coated slow-release technology, reducing loss through leaching and volatilization and maintaining higher plant nitrogen concentrations over time [45,46]. In contrast, urea releases nitrogen rapidly, leading to short-term surplus and subsequent loss, resulting in nitrogen deficiency in later growth stages and higher dilution rates [47,48]. Across harvests, parameter a decreased with successive cuts, likely due to physiological regulation during alfalfa regrowth. The first cut relies on early-applied nitrogen for rapid growth, maintaining high nitrogen concentrations [49]. With each harvest, soil nitrogen supply diminishes, and plants tend to allocate limited nitrogen to stress-resistance organs rather than aboveground regrowth, reducing nitrogen concentrations [50,51]. Cutting stress may also induce secondary metabolites, such as phenolics and lignin, consuming nitrogen and reducing its availability for biomass synthesis [51]. Notably, the b parameter for the second cut was lower than that of the first and third cuts, suggesting a slower nitrogen dilution rate during the second growth cycle. This may be due to high temperatures and low rainfall in July, which limit nitrogen uptake and biomass accumulation.

### 3.2. Diagnosis of Nitrogen Nutrition in Alfalfa Plants

The NNI model and N_and_ model developed based on the CNDC can accurately quantify nitrogen nutritional changes throughout the crop growth cycle [21]. The NNI reflects the real-time nitrogen status of the plant, while N_and_ captures the long-term balance between nitrogen supply and demand [21]. In this study, NNI showed a decreasing trend over the growth period, while differences in N_and_ values among nitrogen treatments increased progressively (Figure 5 and Figure 7). Similar results were reported by Wang et al. [52] and Ye et al. [53] in their nitrogen diagnosis studies on oat and winter wheat, respectively. This phenomenon is largely attributed to the nitrogen demand patterns across different crop growth stages. As C3 plants, alfalfa, oat, and wheat exhibit rapid shoot growth during vegetative stages and require high nitrogen uptake for organ development. In later stages, nitrogen is increasingly redistributed to storage organs such as roots and root crowns, leading to a reduction in nitrogen concentration in the aboveground biomass [54]. Moreover, environmental factors such as temperature, light, and precipitation during the growing season can indirectly affect nitrogen uptake and allocation, resulting in dynamic changes in plant nitrogen status. High-temperature stress significantly inhibits alfalfa root activity and nitrogen absorption while increasing respiratory metabolism, which accelerates nitrogen assimilation product consumption and leads to internal nitrogen imbalance [55]. In low-temperature conditions, the activity of key enzymes in carbon and nitrogen metabolism and nitrogen transport proteins is suppressed, resulting in reduced nitrogen assimilation and insufficient accumulation. Under prolonged low-light conditions, limited photosynthetic product synthesis can inhibit nitrogen metabolism, causing nitrogen to accumulate in roots and senescing tissues, while its distribution to new leaves is significantly reduced [56]. Water is essential for healthy crop growth. Under drought stress, water deficiency directly suppresses photosynthesis, limits leaf expansion, reduces assimilation area, and lowers dry matter accumulation [57]. Conversely, excessive precipitation can lead to waterlogging, resulting in root zone hypoxia, reduced root respiration, and, with prolonged flooding, root decay. This process also accelerates nitrogen leaching and decreases available nitrogen in the rhizosphere, ultimately undermining plant nitrogen status [58].

The application of exogenous nitrogen fertilizer influences NNI by increasing soil nitrogen supply and reduces N_and_ by meeting plant nitrogen demand. Our results showed that alfalfa NNI increased with nitrogen application rate, while N_and_ showed an opposite trend (Figure 5 and Figure 7). Notably, treatments N2 and C2 exhibited NNI values close to 1 and N_and_ values near 0, suggesting that these fertilizer rates maintained a dynamic balance among nitrogen uptake, assimilation, and allocation. This led to optimized nitrogen use efficiency and helped stabilize soil pH, prevent acidification due to nitrogen transformation, and minimize nitrate leaching and N_2_O emissions [59,60]. Excessive nitrogen application resulted in NNI > 1 and N_and_ < 0, indicating nitrogen accumulation beyond plant demand. This could lead to the buildup of nitrogen degradation products such as free amino acids or ammonia, reducing forage quality. Furthermore, nitrogen status fluctuations varied by fertilizer type. This study found that alfalfa plants treated with CRU exhibited smaller NNI and N_and_ fluctuations (NNI: 0.65~1.25; N_and_: −8.71~24.56) compared to those treated with urea (NNI: 0.62~1.30; N_and_: −8.92~25.69). After urea application, soil nitrogen concentration surged rapidly, causing a short-term increase in NNI. However, as nitrogen was quickly consumed or lost, available nitrogen declined, leading to a subsequent drop in NNI and resulting in pronounced fluctuations. In contrast, CRU application maintained a relatively stable level of available nitrogen in the soil over a longer period, preventing “pulsed” nitrogen uptake and effectively reducing NNI variation.

### 3.3. Yield-Increasing and Nitrogen-Saving Potential of Controlled Release Urea

Urea releases over 80% of its nitrogen within three days after application, and its nitrogen use efficiency (NUE) is generally below 35%. To meet the nitrogen demand of crops throughout the growth cycle, multiple topdressings are typically required [61,62]. In contrast, CRU can sustain nitrogen release over several months, with NUE reaching 50~70% [63]. Based on regression analysis between nitrogen rate and yield for both fertilizer types (Table 2), CRU-treated alfalfa achieved a higher theoretical maximum yield while saving 18.41~20.47% nitrogen input. These findings indicate that CRU offers greater potential for improving yield and reducing nitrogen use compared to conventional urea. The advantages of CRU are largely influenced by fertilizer properties, soil effects, and plant physiological responses. From the perspective of fertilizer characteristics, urea, a fast-acting nitrogen source, provides nutrients rapidly but often leads to nitrogen deficiency in later growth stages. It is also prone to leaching after rainfall or irrigation, resulting in resource waste and groundwater contamination [47,62]. In contrast, CRU minimizes nutrient loss by releasing nitrogen gradually, reducing leaching and volatilization risks, and significantly enhancing NUE [45,46]. Regarding soil effects, long-term urea application can cause nitrogen over accumulation, disrupt the nutrient balance, and destabilize the soil microbial community structure [64]. CRU, on the other hand, helps maintain dynamic nutrient balance, improves soil structure, and enhances water and nutrient retention capacity [65]. From a physiological standpoint, the sustained nitrogen supply from CRU promotes consistent nitrogen uptake by alfalfa roots, enhancing photosynthetic efficiency and improving plant stress resistance [66]. In contrast, the rapid nutrient release of urea may induce excessive vegetative growth, resulting in thin stems, reduced lodging resistance, and ultimately diminished yield and quality [67]. However, CRU also has limitations in practical application. For crops with high nitrogen demand during early growth stages, sole reliance on CRU may not adequately meet nitrogen needs. Additionally, CRU is 3–8 times more expensive than urea, which increases production costs. The polymer coating used for slow release is also difficult to degrade, and its residues may have adverse effects on soil microbial communities and soil structure over time. Therefore, research on the blending effect of CRU with urea and the coating materials of CRU can be carried out in the follow-up so as to improve the economy and universality of the application of CRU.

## 4. Materials and Methods

### 4.1. Experimental Site

The experiment was conducted from April to October in both 2023 and 2024 at the Irrigation Experiment Station of the Jingtai Chuan Electricity-Lift Irrigation Water Resources Utilization Center, Gansu Province (37°23′ N, 104°08′ E; elevation 2028 m). The long-term average annual precipitation, evaporation, sunshine duration, mean radiation, mean air temperature, and frost-free period at the site are 191.6 mm, 2761 mm, 2652 h, 6.18 × 10^5^ J·cm^−2^, 8.5 °C, and 191 days, respectively. The soil at the experimental site is classified as sandy loam, with a bulk density of 1.45 g·cm^−3^, a maximum field water-holding capacity of 24.1% (gravimetric basis), and a pH of 8.11. In the 0–20 cm topsoil layer, the contents of soil organic matter, total nitrogen, total phosphorus, total potassium, available nitrogen, and available phosphorus were 6.09 g·kg^−1^, 1.62 g·kg^−1^, 1.32 g·kg^−1^, 34.03 g·kg^−1^, 74.51 mg·kg^−1^, and 26.31 mg·kg^−1^, respectively. During the alfalfa growing period in 2023, the average precipitation and temperature were 0.51 mm and 20.58 °C, respectively, while in 2024, they were 0.99 mm and 12.28 °C (Figure 9).

### 4.2. Experimental Design

The tested alfalfa cultivar was Longdong purple alfalfa (hereafter referred to as alfalfa). The experiment was arranged with two nitrogen fertilizer types and four nitrogen application rates (pure nitrogen). The fertilizer types included controlled-release nitrogen fertilizer (produced by Shandong Kingenta Ecological Engineering Co., Ltd., N content 30%), and urea (N content 46%). The nitrogen application rates were 0, 80, 160, and 240 kg·ha^−1^ [68], resulting in a total of seven treatments (Table 3), with three replicates per treatment. Alfalfa was initially sown on April 7, 2021, with a row spacing of 30 cm and a seeding rate of 22.5 kg·ha^−1^. Urea was applied in a 6:2:2 ratio at three stages each year: before regrowth of the first cut, after the first cut, and after the second cut. The controlled-release fertilizer was applied once before regrowth of the first cut each year. Phosphorus fertilizer (single superphosphate, P_2_O_5_ content 16%, 50 kg·ha^−1^) and potassium fertilizer (potassium chloride, K_2_O content 60%, 50 kg·ha^−1^) were applied as basal fertilizers before the regrowth of the first cut. Irrigation was conducted using drip tapes, with the irrigation volume set according to the local standard irrigation level for artificial grasslands; the irrigation quota is 75 mm. Drip tapes were installed with a spacing of 60 cm, and each emitter was designed to deliver water at a rate of 2 L·h^−1^. Valves and water meters (accuracy: 0.001 m^3^) were installed on the pipelines to control water volume. Other field management practices followed the local standards for artificial grasslands.

### 4.3. Measurement Items and Methods

#### 4.3.1. Yield

Alfalfa yield was determined using the quadrat sampling method. At the initial flowering stage of each cut, a 1 m^2^ quadrat (1 m × 1 m) with uniform plant growth was selected from each plot. Plants were cut at 5 cm above ground level, and weeds were removed before recording the fresh weight. The samples were inactivated at 105 °C for 0.5 h and then oven-dried at 75 °C to constant weight. After cooling, the dry weight was recorded to calculate the hay yield, which was then converted to yield per unit area (kg·ha^−1^).

#### 4.3.2. Aboveground Biomass and Nitrogen Concentration

Starting from the branching stage of each cut, plant samples were collected at weekly intervals (Table 4). Ten consecutive and uniformly growing alfalfa plants were randomly selected from each plot and harvested. Leaves and stems were separated, inactivated at 105 °C for 30 min, and then oven-dried at 75 °C to a constant weight. After cooling, the dry weights were recorded to calculate aboveground biomass. Dried leaf and stem samples were ground using a small grinder, passed through a 1 mm sieve, and digested using the H_2_SO_4_–H_2_O_2_ method. Total nitrogen content in leaves and stems was determined using a FOSS 2300 fully automatic Kjeldahl nitrogen analyzer. Leaf (or stem) nitrogen accumulation (kg·ha^−1^) = Leaf (or stem) Nitrogen content (%) × Leaf (or stem) dry mass (kg·ha^−1^). The total plant nitrogen accumulation was obtained by summing the nitrogen contents of leaves and stems. Plant nitrogen concentration (%) = Total plant nitrogen accumulation (kg·ha^−1^)/total plant dry mass (kg·ha^−1^) [68].

### 4.4. Model Description

#### 4.4.1. Construction of the Critical Nitrogen Dilution Curve Model

Based on the theoretical framework of the critical nitrogen (N_c_) dilution model proposed by Justes et al. [69], the construction of the curve model involved the following four steps:(1)Analysis of variance (ANOVA) was performed on the aboveground biomass data from each sampling date under different nitrogen treatments. Based on whether alfalfa growth was limited by nitrogen, the treatments were classified into nitrogen-limited (where increasing nitrogen significantly enhanced biomass accumulation) and non-nitrogen-limited groups (where further nitrogen application did not significantly increase biomass accumulation).(2)A linear regression was conducted between aboveground biomass and nitrogen concentration for the nitrogen-limited group.(3)The average aboveground biomass of the non-nitrogen-limited group was taken as the maximum attainable aboveground biomass.(4)For each sampling date, the theoretical critical nitrogen concentration was determined as the y-value (nitrogen concentration) at the intersection point of the linear regression line and a vertical line drawn at the maximum aboveground biomass on the *x*-axis.

The critical nitrogen dilution curve model for alfalfa follows the equation proposed by Greenwood et al. [70]:(1)$$N_{c}=aW^{-b}$$
where *N_c_* is the critical nitrogen concentration (%) of alfalfa, *W* is the maximum dry matter mass of alfalfa (t·ha^−1^), *a* is the nitrogen concentration of the plant when alfalfa reaches 1 t of dry matter mass above ground, and *b* is a statistical parameter for the slope of the dilution curve.

#### 4.4.2. Validation of the Critical Nitrogen Concentration Dilution Curve Model

The commonly used root mean square error (RMSE) and normalized RMSE (n-RMSE) were adopted to evaluate the model accuracy:(2)$$RMSE=\sqrt{\frac{\sum_{i=1}^{n}{\left(s_{i}-m_{i}\right)}^{2}}{n}}$$(3)$$n\text{-}RMSE=\frac{RMSE}{S}\times 100\%$$
where *s_i_* and *m_i_* are the measured and modeled values of critical nitrogen concentration, respectively; *n* is the sample size; and *S* is the mean value of measured concentration. *n-RMSE* < 10%, excellent model stability; 10% < *n-RMSE* < 20%, good model stability; 20% < *n-RMSE* < 30%, fair model stability; *n-RMSE* > 30%, poor model stability [71].

### 4.5. Nitrogen Nutrition Diagnostic Models

#### 4.5.1. Nitrogen Nutrition Index Model

The nitrogen nutrition index (NNI) is the ratio of the actual crop nitrogen concentration to the critical nitrogen concentration, which is used to reflect the nitrogen nutrition of alfalfa plants [39]. It is calculated as:(4)$$NNI=\frac{N_{i}}{N_{c}}$$
where *N_i_* is the measured nitrogen concentration value (%) of alfalfa and *N_c_* is the model simulated nitrogen concentration value (%). When *NNI* = 1, it indicates appropriate nitrogen nutrition of alfalfa plants; *NNI* > 1 indicates excess nitrogen nutrition, and *NNI* < 1 indicates insufficient nitrogen nutrition.

#### 4.5.2. Accumulated Nitrogen Deficit Model

A model for accumulated nitrogen deficit (N_and_) model in alfalfa can be derived from Equation (4):(5)$$N_{and}=N_{cand}-N_{iand}$$
where *N_cand_* is the nitrogen accumulation under the critical nitrogen concentration condition (kg·ha^−1^); *N_iand_* is the actual nitrogen accumulation of alfalfa (kg·ha^−1^). When *N_and_* = 0, alfalfa N nutritional status is appropriate; *N_and_* > 0, alfalfa N accumulation is insufficient; *N_and_* < 0, alfalfa N accumulation is excessive.

#### 4.5.3. Relative Yield

Relative yield (RY) is calculated as the ratio of actual yield under each treatment to the maximum observed yield:(6)$$RY=\frac{Y_{i}}{Y_{max}}$$
where *Y_i_* is the alfalfa yield of each treatment (kg·ha^−1^) and *Y_max_* is the maximum alfalfa yield in each treatment (kg·ha^−1^).

### 4.6. Statistics and Analysis of Data

Microsoft Excel 2016 was used for data organization and calculations. SPSS 2.1.0 was used for one-way ANOVA and two-way ANOVA, and the least significant difference (LSD) method was used for comparisons between treatments. Origin 2021 was used for curve fitting and graphing. The level of significance was set at *p* = 0.05.

## 5. Conclusions

(1)The relationship between aboveground biomass and nitrogen concentration in alfalfa followed a power function. The critical nitrogen concentration dilution curve (CNDC) models constructed for the application of urea and controlled-release urea (CRU) to alfalfa all have R^2^ values > 0.99 and normalized root mean square errors (n-RMSE) ranging from 3.1~13%, indicating strong model reliability.(2)The nitrogen nutrition index (NNI) and cumulative nitrogen deficit (N_and_) models constructed based on the CNDC effectively diagnosed nitrogen nutritional status in alfalfa, with consistent results between the two models. Treatments N2 and C2 were identified as optimal in maintaining nitrogen balance and crop performance.(3)Compared to urea, CRU application increased alfalfa yield by 6.60~23.19%. The optimal nitrogen application rates were 175.44~181.71 kg·ha^−1^ for urea and 145.63~153.40 kg·ha^−1^ for CRU. At theoretical maximum yield, CRU saved 18.41~20.47% of nitrogen input relative to urea.

In summary, this study established a reliable nitrogen nutrition diagnostic model for alfalfa and quantified that the application of controlled-released urea can significantly reduce the nitrogen fertilizer rate while achieving high alfalfa yields compared to urea. This finding is of great significance for optimizing nitrogen fertilizer management in alfalfa production and promoting the green and sustainable development of the alfalfa industry. It also provides a valuable reference for the widespread application of controlled-released urea in alfalfa production.

## Figures and Tables

**Figure 1 plants-14-01782-f001:**
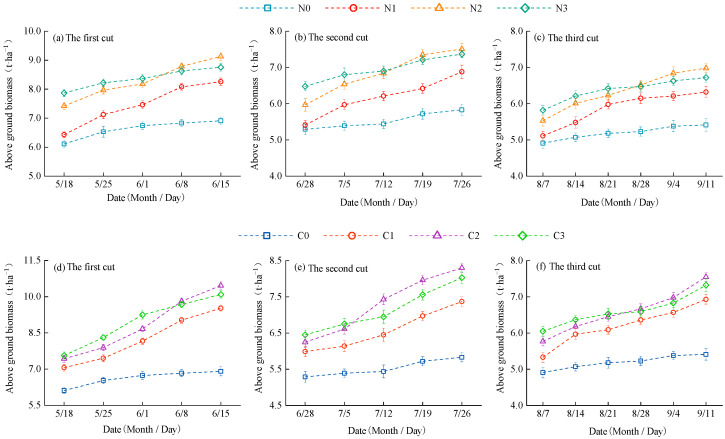
The dynamic accumulation of aboveground biomass of alfalfa. (**a**–**c**) show changes in aboveground biomass of alfalfa plants with urea application; (**d**–**f**) show changes in the aboveground biomass of alfalfa plants with CRU application.

**Figure 2 plants-14-01782-f002:**
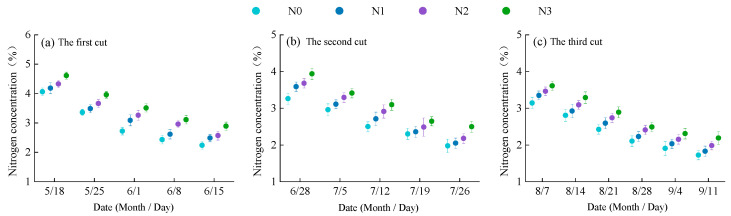
Changes in nitrogen concentration in alfalfa plants at different stages of growth. (**a**–**c**) show the changes in nitrogen concentration of alfalfa plants with urea application; (**d**–**f**) show the changes in nitrogen concentration of alfalfa plants with CRU application.

**Figure 3 plants-14-01782-f003:**
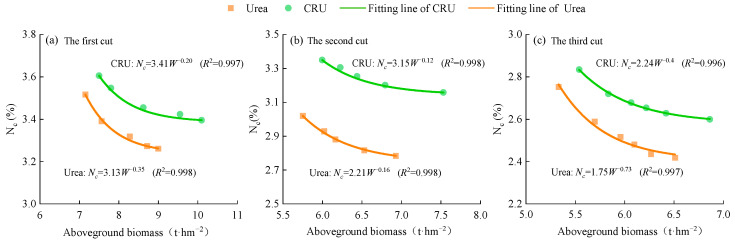
Nitrogen concentration dilution curves in aboveground biomass of alfalfa.

**Figure 4 plants-14-01782-f004:**
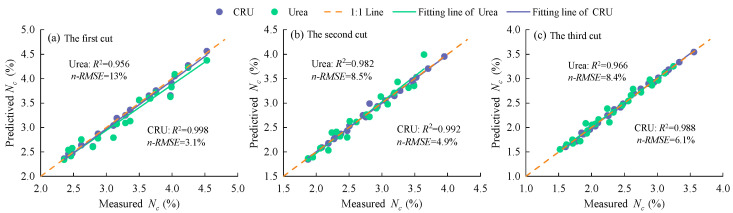
The 1:1 relationship diagram of simulated and observed values of critical N concentration.

**Figure 5 plants-14-01782-f005:**
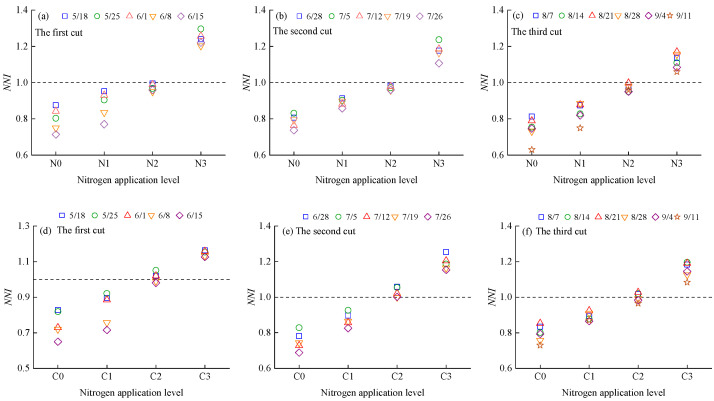
Dynamic changes in the NNI of alfalfa plants under different nitrogen fertilizer management. (**a**–**c**) show the changes in NNI of alfalfa plants with urea application; (**d**–**f**) show the changes in NNI of alfalfa plants with CRU application.

**Figure 6 plants-14-01782-f006:**
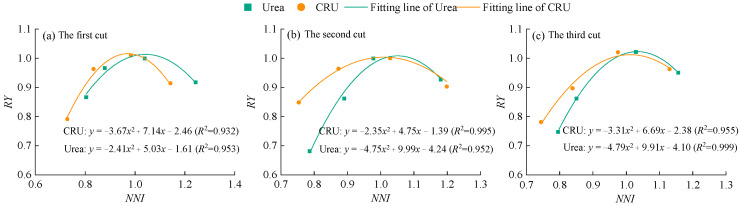
The relationship between NNI and RY of alfalfa.

**Figure 7 plants-14-01782-f007:**
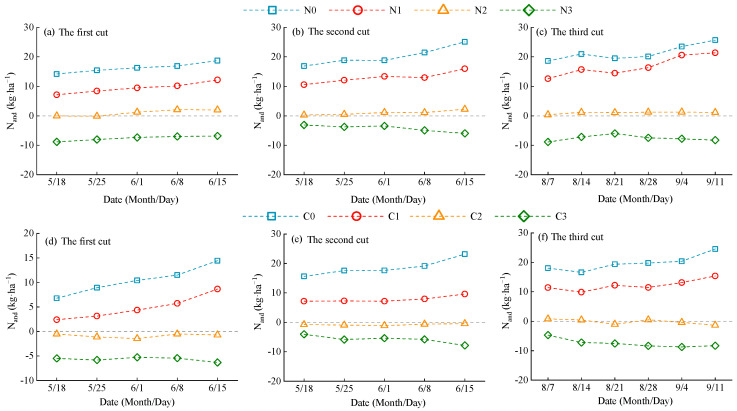
The dynamic changes in N_and_ of alfalfa plants. (**a**–**c**) show the changes in N_and_ of alfalfa plants with urea application; (**d**–**f**) show the changes in N_and_ of alfalfa plants with CRU application.

**Figure 8 plants-14-01782-f008:**
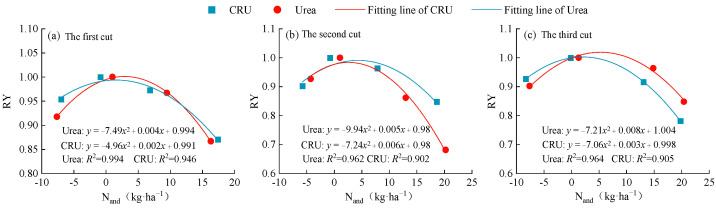
Relationship between N_and_ and RY of alfalfa.

**Figure 9 plants-14-01782-f009:**
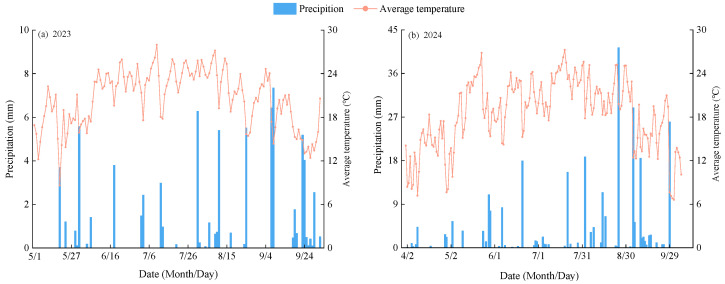
Daily precipitation and air temperature during the alfalfa growing period.

**Table 1 plants-14-01782-t001:** Effect of nitrogen fertilizer management on nitrogen concentration in alfalfa plants.

Nitrogen Fertilizer Type	Nitrogen Application Rate (kg·ha^−1^)	The First Cut (%)	The Second Cut (%)	The Third Cut (%)
Urea	N0	2.96 e	2.59 d	2.35 d
	N1	3.17 d	2.76 cd	2.49 cd
	N2	3.35 cd	2.90 c	2.63 cb
	N3	3.61 ab	3.11 b	2.79 ab
CRU	C0	2.96 e	2.59 d	2.35 d
	C1	3.23 d	2.87 c	2.53 cd
	C2	3.49 bc	3.13 b	2.72 b
	C3	3.79 a	3.36 a	2.94 a

Different lowercase letters after the data indicate significant differences (*p* < 0.05).

**Table 2 plants-14-01782-t002:** Functional relationships between nitrogen fertilizer management and alfalfa yield.

Year	Nitrogen Fertilizer Type	Regression Equation	*R* ^2^	Maximum Yield (t·ha^−1^)	Optimum Nitrogen Rate (kg·ha^−1^)
2023	Urea	y = −9.96 × 10^−5^x^2^ + 0.04x + 11.47	0.925	14.76	181.71
	CRU	y = −2.22 × 10^−4^x^2^ + 0.07x + 11.54	0.987	16.76	153.46
2024	Urea	y = −1.14 × 10^−4^x^2^ + 0.04x + 13.89	0.846	17.40	175.44
	CRU	y = −3.09 × 10^−4^x^2^ + 0.09x + 14.11	0.998	20.66	145.63

**Table 3 plants-14-01782-t003:** Nitrogen fertilizer management.

Nitrogen Type	Nitrogen Application Rate (kg·ha^−1^)
Urea	0 (N0)
	80 (N1)
	160 (N2)
	240 (N3)
CRU	0 (C0)
	80 (C1)
	160 (C2)
	240 (C3)

**Table 4 plants-14-01782-t004:** Sampling dates (day/month).

Year	The First Cut	The Second Cut	The Third Cut
2023	18 May	28 June	7 August
	25 May	5 July	14 August
	1 June	12 July	21 August
	8 June	19 July	28 August
	15 June	26 July	4 September
	−	−	11 September
2024	16 May	25 June	5 August
	23 May	2 July	12 August
	30 May	9 July	18 August
	6 June	16 July	21 August
	13 June	22 July	2 September
	−	−	9 Septemper

Experimental data from 2023 and 2024 were used for constructing and validating the critical nitrogen dilution curve model for alfalfa, respectively.

## Data Availability

All data supporting this study are included in the article.

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
