# Peer review of "Optimizing Management of Alfalfa (Medicago sativa L.) Nitrogen Fertilizer Based on Critical Nitrogen Concentration Dilution Curve Model"

_plants, 2025, doi:10.3390/plants14121782_

Round 1
Reviewer 1 Report
Comments and Suggestions for Authors
The study focused on the application of critical nitrogen dilution curve (CNDC) model on alfalfa perennial leguminous forage. The results provide sound scientific basis for nitrogen status diagnosis and precision nitrogen application in alfalfa cultivation. Some observations are given below:
Line 108: Explain the acronym CRU
Line 114: Move the reference to the table to the beginning of the sentence.
Linev136: Provide a table with LSD (or alternatively the values in brackets) to support the statements. The graph does not show the superiority of CRU over Urea.
Line 405: indicate the volume of the water irrigation per hectare.
Table 3: can you explain to me why in Table 3 there are 5 dates indicated for the first and second cuts while for the third there are 6 dates?
Author Response
We are grateful to all editors and reviewers for their valuable comments and suggestions. These comments are very valuable and helpful for revising and improving our paper, and provide important guidance for our research. We carefully considered every point raised and significantly revised the manuscript with reference to each valuable comment. Furthermore, we would like to show the details as follows:
Comments 1: Line 108: Explain the acronym CRU.
Response 1: Thank you for pointing this out. We agree with this comment. Therefore, we have added the full name of the abbreviation CRU in line 108 of the manuscript and marked it in red in the updated manuscript. The revisions are located on page 3, line 112 of the updated manuscript.
Comments 2: Line 114: Move the reference to the table to the beginning of the sentence.
Response 2: Thank you for your valuable feedback. We have revised the manuscript to place the citation for Figure 1 at the beginning of the sentence. The revisions are on page 3, line 119 of the updated manuscript, and marked it in red in the updated manuscript.
Comments 3: Linev136: Provide a table with LSD (or alternatively the values in brackets) to support the statements. The graph does not show the superiority of CRU over Urea.
Response 3: Thank you very much for pointing out the shortcomings of the data presentation in the manuscript, your comments are very important to improve the quality of the manuscript. According to your suggestion, we have presented the plant N concentration of different N fertilizer delivery in the form of a table in Manuscript 2.1.2, and also conducted LSD analysis, in order to present the results of the study more clearly. The revisions are highlighted in red in the updated manuscript. The revisions are on page 4, line 140, and page 5, line 153, of the updated manuscript. The revised manuscript is excerpted below, where the black bolded parts are the revised contents.
The nitrogen concentration in alfalfa plants showed a decreasing trend as the growth period progressed (Figure 2), indicating a nitrogen dilution effect during biomass accumulation. With increasing nitrogen application rates, the nitrogen concentration increased correspondingly. At the same nitrogen rate, CRU resulted in higher nitrogen concentrations in alfalfa plants compared to urea (Table 1). Under urea treatment, nitrogen concentrations across the three cuts ranged from 2.24%~4.61% (Figure 2a), 1.97%~3.93% (Figure 2b), and 1.72%~3.61% (Figure 2c), respectively. Significant differences were observed be-tween N2 and N0, N3 and N0, and N3 and N1. Under CRU treatment, nitrogen concentrations ranged from 2.24%~4.62% (Figure 2d), 1.97%~4.16% (Figure 2e), and 1.72%~3.83% (Figure 2f), respectively. Significant differences were found between C2 and C0, C3 and C0, and C3 and C1.
Table 1. Effect of nitrogen fertilizer management on nitrogen concentration in alfalfa plants
|
Nitrogen fertilizer type |
Nitrogen fertilizer rate (kg·ha⁻¹) |
The first cut (%) |
The second cut (%) |
The third cut (%) |
|
Urea |
N0 |
2.96e |
2.59d |
2.35d |
|
N1 |
3.17d |
2.76cd |
2.49cd |
|
|
N2 |
3.35cd |
2.90c |
2.63cb |
|
|
N3 |
3.61ab |
3.11b |
2.79ab |
|
|
CRU |
C0 |
2.96e |
2.59d |
2.35d |
|
C1 |
3.23d |
2.87c |
2.53cd |
|
|
C2 |
3.49bc |
3.13b |
2.72b |
|
|
C3 |
3.79a |
3.36a |
2.94a |
Comments 4: Line 405: indicate the volume of the water irrigation per hectare.
Response 4: Thanks for your positive comments. We have added a note on irrigation quotas in the Materials and Methods section to ensure a more rigorous experimental design. The revisions are on page 12, line 405-406 of the updated manuscript, which we have highlighted in red.
Comments 5: Table 3: can you explain to me why in Table 3 there are 5 dates indicated for the first and second cuts while for the third there are 6 dates?
Response 5: Thank you very much for raising this issue, which is very helpful in improving the quality of our manuscripts. Inconsistent sampling frequencies can indeed cause confusion for readers. Below is my response to this issue. This study primarily monitors the nitrogen nutrition status of alfalfa plants from the branching stage to the initial flowering stage. The third crop of alfalfa is in its growth phase during late summer and early autumn. Due to climatic factors and the impact of harvesting the first two crops, the growth process of the third crop is slower than the first two, and its growth period is longer than the first two. Therefore, increasing the number of sampling occasions during the growth process of the third crop of alfalfa is to ensure a more accurate diagnosis of the nitrogen nutrition status of the plants.

Reviewer 2 Report
Comments and Suggestions for Authors
Located in the northwestern part of China, Gansu Province benefits from high solar radiation and significant daily temperature fluctuations, creating ideal conditions for growing alfalfa. In recent years, Gansu has emerged as a key hub for producing high-quality alfalfa, representing over 60% of China’s commercial cultivation area. Nonetheless, challenges like an arid climate and poor soil quality restrict its production capacity.
Against this backdrop, the current study conducted a comprehensive analysis of nitrogen accumulation patterns in alfalfa across various harvests, focusing on the effects of applied urea and controlled-release nitrogen fertilizers. The study aimed to: (1) measure differences in biomass production and plant nitrogen content between the two fertilizer types and develop CNDC models; (2) establish nitrogen nutrition index (NNI) and cumulative nitrogen deficit (Nand) models derived from CNDC to identify nitrogen excess or deficiency thresholds throughout the growing season; and (3) assess the potential of controlled-release urea (CRU) to enhance yields and conserve nitrogen in arid regions of Northwest China, thereby providing a scientific foundation for precision nitrogen management in alfalfa cultivation.
As conclusions and summaries, the authors stated:
>The connection between aboveground biomass and nitrogen content in alfalfa can be described using a power function. The critical nitrogen dilution curves (CNDCs) established under urea and controlled-release urea (CRU) treatments demonstrated excellent fit to the data, with R² values exceeding 0.99 and normalized root mean square errors (n-RMSE) ranging from 3.1% to 13%. These results reflect high model accuracy and reliability.
>The nitrogen nutrition index (NNI) and the cumulative nitrogen deficit (Nand) models, both based on the CNDCs, proved effective in assessing the nitrogen nutritional status of alfalfa. The models produced consistent findings, with treatments N2 and C2 identified as optimal for maintaining nitrogen balance and maximizing crop performance.
>When compared to urea, the application of CRU increased alfalfa yield by between 6.60% and 23.19%. The best nitrogen application rates were approximately 175.44 to 181.71 kg·ha⁻¹ for urea, and 145.63 to 153.40 kg·ha⁻¹ for CRU. At the point of maximum theoretical yield, using CRU resulted in an 18.41% to 20.47% reduction in nitrogen input compared to urea.
It is true that Ms is comprehensive, containing numerous results, and may not be immediately easy to interpret. Nevertheless, the paper is well structured and acceptable. I have a few comments:
- Why is Figure 10 placed in the Materials and Methods section? It would be better if this figure, along with the two tables, were moved to the Results section in an appropriate location to clearly illustrate the connection between your findings.
- The quality of the figures is quite poor; please work on enhancing their clarity and presentation.
- Try to condense the Results section. There are several instances of repeated percentages; instead, you can refer to your tables to avoid redundancy.
- Additionally, consider comparing your findings with similar studies in the existing literature.
- In the conclusion, it is important to explicitly highlight the novelty and significance of your research results.
- The literature review should be expanded. Incorporate 4 to 6 additional relevant studies from this journal, while keeping self-citations to a minimum.
Author Response
We are grateful to all editors and reviewers for their valuable comments and suggestions. These comments are very valuable and helpful for revising and improving our paper, and provide important guidance for our research. We carefully considered every point raised and significantly revised the manuscript with reference to each valuable comment. Furthermore, we would like to show the details as follows:
Comments 1: Why is Figure 10 placed in the Materials and Methods section? It would be better if this figure, along with the two tables, were moved to the Results section in an appropriate location to clearly illustrate the connection between your findings.
Response 1: Thank you very much for your valuable time to review our submitted manuscript, your comments are very professional and open new ideas for our research and paper writing. In our initial manuscript, Figure 10, Tables 2 and 3 mainly present the basic conditions of the experimental site and the experimental design. Due to the requirements of the journal layout, the Materials and Methods are located after the Discussion, and it is not too intuitive to cite them in the Results section, therefore, we have cited Figure 10 and Tables 2 and 3 in the Materials and Methods. We apologize that such a citation caused confusion in your reading, and we will consider your suggestion first in our subsequent research and paper writing. Thank you again for your suggestions.
Comments 2: The quality of the figures is quite poor; please work on enhancing their clarity and presentation.
Response 2: Thank you most sincerely for your review. We understand the importance of chart quality in communicating information, and we have done our best to optimize all the images in the manuscript in terms of data accuracy, visual presentation, and design specifications. In the updated manuscript, we have uploaded a separate attachment containing all the images in the manuscript for your review.
Comments 3: Try to condense the Results section. There are several instances of repeated percentages; instead, you can refer to your tables to avoid redundancy.
Response 3: Thank you very much for your expert comments on the manuscript, your comments have been very profitable. Indeed, in our first submission of the manuscript, there was redundant content in the results, so we have streamlined the results of 2.3.2 and 2.4 by removing the redundant and repetitive percentage data to improve the conciseness and readability of the results. The revised manuscript is excerpted below, where the black bolded parts are the revised contents. The revisions are on page 7, line 222, and page 8, line 237-242, of the updated manuscript, which we have highlighted in red.
Comments 4: Additionally, consider comparing your findings with similar studies in the existing literature.
Response 4: Thank you for taking the time to provide professional feedback on the manuscript. Your comments have helped me to further refine the research content. Based on your suggestions, we have added a comparison between this study and previous research in Section 3.1. This provides readers with a broader contextual understanding and helps to validate the reliability and significance of our research findings. The revisions are on page 9, lines 263-270 of the updated manuscript, where we have highlighted them in red. The revised manuscript is excerpted below, where the black bolded parts are the revised contents.
The essence of nitrogen dilution in crops lies in the fact that the rate of biomass accumulation exceeds the rate of nitrogen uptake or supply, resulting in a decline in nitrogen concentration per unit biomass [36]. The use of the CNDC model to diagnose crop nitrogen status has been widely applied to various cereal and cash crops [37-39]. In the CNDC model, parameter a represents the nitrogen concentration when the aboveground biomass reaches 1 t·ha⁻¹, while b represents the rate at which nitrogen concentration decreases with increasing biomass [35,40]. In this study, we developed CNDC models for alfalfa based on aboveground biomass under urea and CRU treatments. The fitted equations were as follows: Urea: Nc = 3.13W⁻0.35, Nc = 2.21W⁻0.16, Nc = 1.75W⁻0.73, CRU: Nc = 3.41W⁻0.20, Nc = 3.15W⁻0.12, Nc = 2.24W⁻0.40 (Figure 3). All parameter a values were lower than those reported by Lemaire et al. [41] for alfalfa (Nc = 4.8W⁻0.34), likely due to differences in climatic and soil nutrient conditions. Lemaire’s [41] experiments were conducted in France under mild and humid conditions with evenly distributed rainfall and favorable soil hydrothermal environments, which promoted root vigor, nitrogen uptake, and carbon-nitrogen metabolism in alfalfa. In contrast, our experiment was conducted in the arid climate of Northwest China, where low precipitation and frequent water stress reduced photosynthetic efficiency and carbon-nitrogen assimilation, resulting in lower nitrogen concentrations in alfalfa plants. However, Chen et al. [42] constructed a critical nitrogen concentration model for alfalfa is Nc = 3.352W0.391 (Aohan alfalfa) and Nc = 2.673W0.485 (Gongnong No.1 alfalfa), the model parameters of Chen et al. study differ significantly from those of the critical nitrogen con-centration dilution curve model constructed in this study. It may be due to differences in the research objectives and the composition of above-ground biomass. Chen et al. [42] focused on seed-producing alfalfa, with the aboveground biomass of alfalfa including leaves, stems, flowers, pods, and seeds. This study focused on forage alfalfa, with the aboveground biomass of alfalfa comprising only leaves and stems.
Comments 5: In the conclusion, it is important to explicitly highlight the novelty and significance of your research results.
Response 5: Thank you for your meticulous review of the manuscript. Your comments have helped to improve my paper. Emphasizing the novelty and significance of the research results will enable readers to clearly recognize the unique value of the study. In our previous submission, the summary of the conclusions was incomplete, and the innovation and significance of the paper were not sufficiently highlighted. Therefore, based on your unique insights, we have revised the content of the conclusions to better emphasize the significance and innovation of the paper. The revisions are on page 15, line 500 and page 15, lines 512-518 of the updated manuscript, we have marked it in red. The revised manuscript is excerpted below, where the black bolded parts are the revised contents.
(1) The relationship between aboveground biomass and nitrogen concentration in alfalfa followed a power function. The critical nitrogen concentration dilution curve (CNDC) models constructed for the application of urea and controlled-release nitrogen fertilizers (CRU) to alfalfa all have R² values > 0.99, and normalized root mean square errors (n-RMSE) ranging from 3.1%~13%, indicating strong model reliability.
(2) The nitrogen nutrition index (NNI) and cumulative nitrogen deficit (Nand) models constructed based on the CNDC effectively diagnosed nitrogen nutritional status in alfalfa, with consistent results between the two models. Treatments N2 and C2 were identified as optimal in maintaining nitrogen balance and crop performance.
(3) Compared to urea, CRU application increased alfalfa yield by 6.60%~23.19%. The optimal nitrogen application rates were 175.44~181.71 kg·ha⁻¹ for urea and 145.63~153.40 kg·ha⁻¹ for CRU. At theoretical maximum yield, CRU saved 18.41%~20.47% of nitrogen input relative to urea.
In summary, this study established a reliable alfalfa nitrogen nutrition diagnosis model system and quantified that the application of controlled-release nitrogen fertilizer can significantly reduce nitrogen fertilizer input while achieving high alfalfa yields. This finding has important practical implications for optimizing alfalfa nitrogen fertilizer management, reducing production costs, minimizing environmental pollution risks, and promoting the green and sustainable development of the alfalfa industry.
Comments 6: The literature review should be expanded. Incorporate 4 to 6 additional relevant studies from this journal, while keeping self-citations to a minimum.
Response 6: We sincerely appreciate your valuable comments. We have expanded the discussion in the introduction, the revisions are on page 2, lines 70-73, page 2, lines 78-82, and page 2, lines 86-88 of the updated manuscript, which we have highlighted in red. At the same time, we have scrutinized the literature and carefully read many related studies in this journal, which can provide very useful references for this study, therefore, we have added more related studies in this journal in the introduction section of the revised manuscript. The specific references added are on page 16, line 542, page 16, line 553, page 16, line 554, page 16, line 571, and page 16, line 577. The added references are excerpted below.
4. Zhao, Y.; Wang, Y.Q.; Sun, S.N.; Liu, W.T.; Zhu, L.; Yan, X.B. Different Forms and Proportions of Exogenous Nitrogen Promote the Growth of Alfalfa by Increasing Soil Enzyme Activity. Plants 2022, 11, 1057.
9. Barłóg, P. Improving fertilizer use efficiency—Methods and strategies for the future. Plants 2023, 12, 3658.
10. Mishra, S.; Levengood, H.; Fan, J.P.; Zhang, C.K. Plants Under Stress: Exploring Physiological and Molecular Responses to Nitrogen and Phosphorus Deficiency. Plants 2024, 13, 3144.
11. Liu, C.; Liu, J.J.; Wang, J.; Ding, X.Y. Effects of Short-Term Nitrogen Additions on Biomass and Soil Phytochemical Cycling in Alpine Grasslands of Tianshan, China. Plants 2024, 13, 1103.
20. Lei, H.J.; Fan, Y.M.; Xiao, Z.Y.; Jin, C.C.; Chen, Y.Y.; Pan, H.W. Comprehensive evaluation of tomato growth status under aerated drip irrigation based on critical nitrogen concentration and nitrogen nutrient diagnosis. Plants 2024, 13, 270.

Round 2
Reviewer 2 Report
Comments and Suggestions for Authors
It is acceptable, now.